# Women’s Self-Assessment of Quality of Life and Menopausal Symptoms: An Online Survey of 26,000 Women in German-Speaking Countries

**DOI:** 10.3390/ijerph22101502

**Published:** 2025-09-30

**Authors:** Olivier Flückiger, Alexander Krannich, Peter Recknagel, Markus Leiter, Tamara Stix-Steinwald, Eva Poggio, Christoph Hillen, Irit Nachtigall

**Affiliations:** 1RoX Health GmbH, 10997 Berlin, Germany; olivier@roxhealth.com (O.F.); markus@roxhealth.com (M.L.); tamarastix@gmail.com (T.S.-S.); eva.poggio@gmail.com (E.P.); christoph@reesi.de (C.H.); 2BioStats GmbH, 14641 Nauen, Germany; alexander.krannich@biostats.de; 3Scientific Consulting in Life Science, 98529 Suhl, Germany; 4Faculty of Medicine, MSB Medical School Berlin, University of Health and Medicine, Rüdesheimer Straße 50, 14197 Berlin, Germany; irit.nachtigall@vivantes.de; 5Vivantes Network for Health, 13407 Berlin, Germany

**Keywords:** menopause, quality of life, MRS, sleep, online survey, DACH, insomnia, women’s health

## Abstract

Menopausal symptoms can substantially impair women’s quality of life, yet large-scale, population-based data from German-speaking regions are lacking. We used data of a cross-sectional online survey among 26,338 women in Germany, Austria, and Switzerland, using validated instruments such as the Menopause Rating Scale (MRS II), Insomnia Severity Index (ISI), and Alcohol Use Disorders Identification Test (AUDIT-C). Additional questions covered weight change, employment, nutrition, and physical activity. We investigated correlations and group differences using descriptive statistics and univariate tests. The average MRS score was 16.94, with 51% classified as severe discomfort and 15% as requiring treatment. Insomnia was common, with 90.3% reporting at least some degree of sleep disturbance and nearly half (48%) meeting criteria for clinical insomnia. Sleep problems were significantly associated with menopausal symptom severity. No correlations were found between MRS scores and reported nutrition and exercise. Unemployment and weight gain were significantly associated with higher symptom burdens. These findings highlight the considerable health burden faced by menopausal women in the DACH region and suggest a substantial unmet need for clinical and public health interventions.

## 1. Introduction

Menopause is a natural biological transition defined by the permanent cessation of menstruation, typically occurring between the ages of 45 and 55 years, with the average age of onset around 51 years in Western countries [1,2]. It marks the end of a woman’s reproductive capacity and is diagnosed retrospectively after 12 consecutive months of amenorrhea in the absence of other pathological or physiological causes [3]. Globally, an estimated 1.2 billion women will be postmenopausal by 2030 [4], underscoring the importance of this transitional life stage in public health and clinical medicine.

The menopausal transition involves a complex interaction of biological, genetic, and psychological factors and is characterized by a marked decline in circulating estrogen levels. This hormonal shift, rather than the loss of ovulatory capacity per se, disrupts multiple physiological systems and can manifest in a broad spectrum of symptoms. [5,6]. Most common symptoms range from hot flashes and sleep disturbances to urogenital problems, mood-related symptoms such as anxiety, depression, irritability, as well as sexual dysfunction and musculoskeletal symptoms, including joint pain, stiffness, and an increased risk of osteoporosis [7,8].

These multifactorial symptoms often cluster and interact, creating a compounded burden on women’s physical, mental, and sexual health, significantly impacting their quality of life (QoL). As women now live for approximately one-third of their lives after menopause, addressing these challenges is essential for promoting healthier aging and well-being throughout the lifespan.

Despite the availability of hormone replacement therapy (HRT) as an effective option for many symptoms, concerns about its risks (e.g., cardiovascular events, breast cancer) [9,10] have led to increased interest in non-pharmacological strategies such as exercise [11], complementary therapies, and lifestyle interventions [12,13,14].

The Menopause Rating Scale II (MRS II) is a validated instrument widely used in clinical and research settings to assess the severity of these symptoms. While the prevalence and profile of menopausal symptoms have been described in several countries, comprehensive, large-scale data for German-speaking populations remain limited. Cultural, healthcare system, and lifestyle differences may also influence how symptoms are perceived and reported.

This investigation aims to fill that gap by presenting findings from an online survey from more than 26,000 women in Germany, Austria, and Switzerland. We evaluate the burden of menopausal symptoms using MRS II and examine their relationship with sleep quality, alcohol use, employment status, weight change, nutrition, and physical activity. Given its widespread prevalence and impact on work productivity, mental health, and healthcare needs, menopause represents a substantial yet under-recognized public health and socioeconomic issue. Given the scale of affected individuals and the long-lasting impact on women’s daily functioning, menopause should be recognized as a pressing public health issue requiring structured attention beyond individual clinical care. Therefore, findings should offer important insights for clinicians, researchers, and policymakers into the lived experiences of midlife women and may inform more targeted interventions.

## 2. Materials and Methods

### 2.1. Study Design and Recruitment

This was a cross-sectional, observational data collection based on a large commercial online survey conducted between 2023 and 2024. Participation was open to adult women aged 18 years and older residing in the DACH region (Germany, Austria, and Switzerland). Recruitment was carried out via convenience sampling through relevant social media channels, using a self-administered online questionnaire without interviewer involvement. Participants received individualized feedback after data validation. The survey was self-administered, anonymous, and conducted in German. We used the anonymized collected data retrospectively for our research.

### 2.2. Inclusion and Exclusion Criteria

Eligible participants were biologically female and at least 18 years old. No exclusions were made based on menopausal status, current treatment, or health conditions. The survey was openly advertised as part of a broader initiative on menopause health, primarily via social media channels.

### 2.3. Survey Instruments

The online questionnaire comprised a range of standardized and validated instruments to assess menopausal symptoms and related health behaviors. Core symptom burden was measured using the Menopause Rating Scale II (MRS II) [15], a well-established tool that evaluates 11 symptoms across three domains: somato-vegetative (e.g., hot flashes, sleep disturbances), psychological (e.g., mood changes, irritability), and urogenital (e.g., vaginal dryness, bladder dysfunction). Each item is scored on a 5-point Likert scale ranging from 0 (“not at all”) to 4 (“very severe”), allowing for both domain-specific and total symptom scoring. According to established thresholds proposed by Blümel et al., a total MRS score of ≥14 indicates the need for treatment for climacteric symptoms, while a score of ≥17 reflects severe discomfort and may signal more intense clinical intervention needs [16].

To assess sleep quality and insomnia symptoms, the Insomnia Severity Index (ISI) was employed [17]. This seven-item instrument captures the nature, severity, and impact of sleep difficulties and classifies respondents into four severity categories: 0–7 indicates no clinically significant insomnia, 8–14 reflects subthreshold insomnia, 15–21 corresponds to moderate insomnia, and 22–28 indicates severe insomnia.

Alcohol consumption patterns were evaluated using the AUDIT-C questions Q1 (alcohol frequency) and Q2 (alcohol quantity) [18], a brief screening tool consisting of three items that measure the frequency and quantity of alcohol intake.

Lifestyle and dietary behavior were assessed using adapted items based on the “10 Rules of the German Nutrition Society (DGE)” [https://www.dge.de/fileadmin/dok/english/10-guidelines/10-guidelines-wholesome-diet-dge.pdf, accessed on 23 July 2025], covering aspects such as intake of fruits and vegetables, whole grains, hydration, and physical activity.

Additional self-reported data included employment status, weight change (categorized as gain, loss, or stable), and basic sociodemographic characteristics such as age, education, and region of residence.

### 2.4. Data Privacy and Ethics

Data collection complied with GDPR standards. Participation was voluntary with consent given electronically before the survey began. The data were fully collected in anonymized form.

### 2.5. Statistical Analysis

Analyses were conducted using R software (version: 4.5.1). Descriptive statistics summarized participant characteristics and instrument scores. Nonparametric tests (e.g., Kruskal–Wallis rank sum test) were used for group comparisons. Correlation analyses (Spearman rank correlation) evaluated associations between symptoms and other variables. Missing values were handled using complete case analysis; no imputation was performed. Descriptive results are presented as means with standard deviations or as percentages, as appropriate. A *p* < 0.05 was considered significant. All *p*-values are considered as explorative and non-confirmatory, due to the observational design.

### 2.6. Use of GenAI Tools

Artificial intelligence tools (e.g., ChatGPT-5, OpenAI, San Francisco, CA, USA) were used to assist in language refinement and manuscript editing. The use was in accordance with WAME Recommendations on Chatbots and Generative Artificial Intelligence in Relation to Scholarly Publications (https://wame.org/page3.php?id=106, accessed on 19 June 2025). All scientific content and conclusions are the authors’ own.

## 3. Results

### 3.1. Baseline Characteristics

A total of 26,338 women participated in the study. The mean age was 47.9 years (SD 4.3, range from 25 to 75 years). The average BMI was 28.1 (SD 6.5). The majority of participating women were employed (40% full-time or 42% part-time), although 18% of a sub-cohort reported being unemployed. Of these women, 43% had a regular period, 36% had an irregular period and 21% no longer had a period.

### 3.2. Menopausal Symptoms (MRS II)

The average total MRS II score was 16.94 (SD 7.45). Based on established cutoffs [16], 15% of participants met criteria for “need for treatment” (MRS ≥ 14), while 51% reported “severe discomfort” (MRS ≥ 17). Psychological symptoms such as irritability (mean 1.90), depressive mood (1.79), and physical/mental exhaustion (2.17) were particularly pronounced. The somato-vegetative domain was also prominently affected, with frequent reports of hot flashes, heart discomfort, and sleep problems. The urogenital domain, while scoring lower overall, still revealed substantial discomfort in a notable proportion of women, particularly related to sexual problems and bladder complaints (Table 1). These symptoms also showed high inter-item correlations, suggesting clustered distress in all reported domains. There was no correlation between age and MRS (r = 0.04). The majority of participants were aged between 40 and 65 years, with only 36 respondents under 40 and 7 over 65. Correlation within MRS items and with age is shown in Figure 1.

Women with regular periods had a mean MRS II score of 15.4 (mean age 46.2 years), those with irregular periods 17.5 (47.9 years), and those without periods 19.0 (51.3 years).

### 3.3. Sleep Disturbances (ISI)

Among 4058 women who completed the ISI, the average score was 14.16 (SD 4.87). Only 9.7% of participants reported no insomnia, while moderate to severe insomnia affected 48.3% of this subsample. Correlational analyses revealed that ISI scores were associated with multiple MRS domains, particularly psychological and somato-vegetative symptoms (Figure 1, Table 2 and Figure 2).

The ISI score differed slightly with a mean value of 13.7 for women with a regular period, 14.3 with an irregular period and 14.8 with no period.

### 3.4. Alcohol Consumption (AUDIT-C)

The mean composite AUDIT-C score was 2.24 (SD 1.20). Despite alcohol use being common, no significant correlation was found between alcohol scores and symptom burden (MRS), suggesting minimal direct impact within this cohort.

### 3.5. Nutrition and Physical Activity

Indicators of diet quality and physical activity based on the DGE 10-rule framework were not associated with symptom burden. Even though many respondents reported suboptimal behavior (e.g., low fruit/vegetable intake), no relevant correlations to MRS scores were observed.

### 3.6. Employment Status and Weight Change

In a subsample of 1606 women with employment data, unemployed women had significantly higher MRS (mean 20.5 vs. 16.8–17.9) and ISI scores (mean 16.3 vs. 13.9–14.3) compared to employed peers (*p* < 0.001). Additionally, 54% of respondents reported weight gain in recent years, and this group showed a trend toward higher symptom scores, underscoring a potential link between perceived body changes and menopausal distress. Furthermore, weight gain and weight loss were considered together with the MRS II. As can be seen in Figure 3, there are patterns indicating that weight stability is associated with a lower MRS II.

## 4. Discussion

This large-scale, survey-based, cross-sectional study set out to address a significant knowledge gap by presenting findings from an online survey of over 26,000 women residing in Germany, Austria, and Switzerland. Using the validated Menopause Rating Scale II (MRS II), we assessed the burden of menopausal symptoms and explored their associations with sleep quality, alcohol consumption, employment status, weight change, nutrition, and physical activity. The goal was to generate robust, population-based insights that reflect the lived experiences of midlife women in the DACH region and provide actionable evidence for clinicians, researchers, and policymakers seeking to develop more targeted interventions. The results of our study provide one of the most comprehensive data sets for self-reported symptoms and related quality-of-life factors for menopausal women in the DACH region and underscores the substantial symptom burden that women experience across psychological, somato-vegetative, and urogenital domains.

### 4.1. Menopausal Symptoms (MRS II)

The average MRS II score of 16.94, with more than 50% of participants classified as experiencing “severe discomfort” and 15% meeting the threshold for “need for treatment,” suggests a considerably higher burden than reported in smaller or clinical cohort studies from different countries indicating that previous research may have underestimated the real-world symptom burden in this population [19,20,21,22]. In these studies, severe discomfort as defined by thresholds of 14, 16, or 17 was noted in 15–35% of participants. One study of 427 healthy women reported exceptionally high rates for specific items (95.8% for exhaustion and 95.1% for joint/muscle discomfort) and 88.5% exceeding a treatment-need threshold of ≥14 [16]. In contrast, prevalence rates of severe vasomotor or overall symptoms ranging from 8.1% to 9.6%,have been reported in larger cohorts, although mean scores were not consistently provided [23,24]. All studies employed similar cut-offs (14–16) for treatment need, and variations in reported symptom prevalence appear related to study design, geography, and cultural context.

Psychological symptoms such as irritability, depressive mood, and exhaustion were particularly elevated in our cohort, aligning with international literature that highlights mood disturbances as among the most frequently reported and distressing symptoms during the menopausal transition [25,26,27]. Additionally, only 36 participants were younger than 40 years, and just 7 were older than 65 years, confirming that the age distribution in our sample was narrow. This helps explain the absence of a correlation between age and MRS scores. Average ages by menstrual status (46.2 for regular, 47.9 for irregular, and 51.3 for absent periods) followed the expected menopausal trajectory, but did not show a clear pattern in symptom severity. The high prevalence of severe symptoms in a non-clinical, general population sample highlights an unmet need for scalable public health strategies particularly screening, awareness campaigns, and accessible symptom management tools to address menopause-related quality-of-life decline at the population level.

### 4.2. Sleep Disturbances (ISI)

Sleep problems emerged as a particularly significant contributor to overall symptom burden. Of the women who completed the Insomnia Severity Index (ISI), more than 90% experienced at least some level of insomnia, and almost half reported symptoms in the moderate-to-severe range. These findings are consistent with prior research [28,29] highlighting the strong interplay between hormonal fluctuations, mood regulation, and circadian rhythm disturbances in midlife women [30,31,32]. The observed correlations between ISI scores and both psychological and somato-vegetative symptoms might support a bidirectional relationship. While poor sleep may worsen emotional and somatic complaints, elevated symptom distress can impair sleep quality. Prior studies have demonstrated that insomnia is a mediating factor between menopausal status and quality of life, making it a promising target for intervention [32,33]. Compared to sleep, the status of the period in relation to the MRS II is negligible. Given the high prevalence and strong associations with psychological distress, sleep disturbances during menopause should be recognized as a public health priority, warranting early identification and inclusion in mental health and occupational health programs.

### 4.3. Alcohol Consumption (AUDIT-C)

The average AUDIT-C score of 2.24 suggests that alcohol use was moderate but common in this cohort. Importantly, the lack of a significant association between alcohol consumption and overall MRS score implies that, in this population, drinking behavior did not substantially impact perceived symptom burden. This finding leaves conclusions open and supplements mixed results in the literature, where some studies have found that higher alcohol use is associated with worse vasomotor symptoms or sleep problems while others have found positive or neutral effects depending on drinking patterns, context, and menopausal stage [34,35]. Thus, while alcohol consumption may be relevant in specific symptom subdomains, our data support the interpretation that alcohol does not act as a strong general modulator of menopause-related quality of life in a cross-sectional setting.

### 4.4. Nutrition and Physical Activity

Contrary to common assumptions, lifestyle factors as measured by adherence to healthy dietary practices or physical activity assessed using principles from the German Nutrition Society (DGE) did not show significant associations with symptom severity. While this does not rule out their importance for overall health, it suggests that their role in moderating menopausal symptoms may be less direct or may require more precise measurement tools to detect subtle effects.

This may seem surprising, given consistent findings in clinical and intervention studies suggesting benefits of physical exercise [36,37] and balanced nutrition [38,39,40] on menopausal symptoms, particularly mood, metabolic parameters, and musculoskeletal health. However, the lack of significant associations between lifestyle behaviors and symptom burden in our study should be interpreted cautiously. One possible explanation is the self-reported and relatively simplistic nature of the lifestyle assessment, which may not have been sufficiently sensitive to capture subtle behavioral effects. Additionally, lifestyle-related benefits may require longitudinal tracking to demonstrate symptom reduction, rather than being visible cross-sectionally. Finally, the influence of lifestyle behaviors on menopausal symptoms may be indirect or confounded by psychosocial or socioeconomic factors that were not fully adjusted for. Hence, the absence of a statistical link in this analysis does not necessarily disprove the value of diet and exercise but underscores the need for more refined measurement tools and longitudinal designs. Although no direct associations were found, public health messaging should continue to promote healthy behaviors during midlife, as their indirect benefits on overall aging trajectories remain well-documented. Future initiatives may require more targeted, evidence-based framing to improve adoption and efficacy.

### 4.5. Employment Status and Weight Change

Among women who reported employment data, those who were unemployed exhibited significantly higher MRS and ISI scores, suggesting that socioeconomic instability and loss of daily structure may exacerbate both menopausal symptoms and sleep issues. However, given the cross-sectional nature of the data, it remains unclear whether unemployment contributed to increased symptom severity, or whether more pronounced symptoms may have led to reduced work ability or job loss. This pattern is supported by prior studies linking unemployment and financial stress to greater depressive symptoms, lower coping ability, and reduced healthcare access during menopause [41,42]. Similarly, weight gain as reported by over 50% of respondents was associated with a trend toward higher symptom burden. Though causality cannot be inferred from this cross-sectional design, the literature supports a bidirectional link between menopause, weight gain [43,44], and quality-of-life impairment [45,46]. These findings argue for integrated care approaches that consider not just biological changes but also the social and emotional context in which menopause occurs.

While our main dataset did not include in-depth socio-demographic variables, a smaller subsample of 494 women completed an extended questionnaire that captured several relevant aspects. In this subsample, women reported between 0 and 4 children (median: 2). Regarding residential setting, 25% lived in large cities, 8.7% in urban regions, 34% in small towns, and 32% in rural areas. The vast majority (93%) resided in Germany, with smaller numbers in Austria and Switzerland. Importantly, 95% of respondents reported having at least a small to large social network, while only 5% reported no social connections.

### 4.6. Future Research Directions

To build on these findings, future studies should incorporate longitudinal designs to trace symptom progression, identify predictors of resilience or risk, and evaluate the effectiveness of multimodal interventions. It will also be essential to expand this research to more diverse populations beyond the DACH region and to integrate objective clinical markers (e.g., hormone levels, inflammatory profiles) to complement self-reported outcomes.

In addition, future research would benefit from the inclusion of intersectional analytical frameworks that consider the complex interplay of socioeconomic status, education, employment, family role, cultural attitudes, and access to healthcare. Capturing these dimensions through expanded demographic and psychosocial variables will allow for a deeper understanding of how social determinants shape symptom burden and health outcomes. This may also help identify vulnerable subgroups who require tailored intervention strategies.

Finally, given the high prevalence of sleep and psychological symptoms, dedicated interventional trials targeting these domains may yield significant benefits for symptom relief and functional recovery. Regardless of the causal direction, our findings already highlight an urgent need for integrative, targeted support strategies for midlife women, both to alleviate current symptom burden and to prevent long-term impairments.

### 4.7. Implications for Public Health

The findings of this large-scale, population-based study underscore menopause as a significant but often underrecognized public health concern. With over half of respondents reporting severe symptom burden and a substantial proportion meeting criteria for clinical intervention, it is evident that many women suffer silently without adequate support. Sleep disturbances, mood symptoms, and functional impairment were not restricted to clinical subgroups but were widespread among midlife women in the general population.

These results call for broader public health responses, including education campaigns to destigmatize menopause, systematic integration of menopause-related care into primary health services, and the development of workplace policies that accommodate the needs of affected individuals. Furthermore, the observed links between unemployment, weight gain, and symptom severity highlight the need to address social and structural determinants of health during the menopausal transition.

By framing menopause not merely as a medical condition but as a multidimensional public health issue, these findings provide a foundation for developing holistic, equitable, and scalable interventions that can improve the quality of life for millions of women globally.

### 4.8. Limitations

Several limitations must be acknowledged. First, as with all self-report studies, measurement bias is a concern, particularly given the subjective nature of symptom ratings. However, the use of validated instruments mitigates this risk to some extent but does not eliminate it. Second, the study may be affected by selection bias, as participation was voluntary and recruitment was conducted exclusively online. This may have skewed the sample toward digitally literate, more educated, health-conscious individuals who are more likely to engage with menopause-related content. Third, self-referral bias cannot be ruled out, i.e., women currently experiencing significant distress or experience strong symptoms may have been more motivated to complete the survey, possibly inflating symptom prevalence estimates. Also midlife social stressors, compounded by employment instability or weight concerns, could exacerbate perceived symptom intensity. Furthermore, self-reporting for the assessment of menopause status can be subjective and deviate from medically confirmed diagnoses, which is to be expected in a study with patient-reported outcomes. Finally, the results may have limited generalizability due to the geographic focus on German-speaking countries (DACH region), and caution is warranted when extrapolating findings to other cultural or healthcare contexts. Moreover, the study did not incorporate detailed socio-demographic and psychosocial variables such as educational level, marital status, or cultural attitudes. This limited our ability to conduct intersectional analyses exploring how multiple social determinants of health might interact to influence symptom burden. While a smaller subsample (n = 494) provided more granular information on social support, family structure, and place of residence, the sample size was insufficient for robust stratified analyses. Future research should explicitly address these factors to gain a more nuanced understanding of menopause-related health disparities.

Despite these limitations, the study provides valuable large-scale insight into the lived experiences of menopausal women and highlights key areas for clinical attention.

## 5. Conclusions

This study reveals a high prevalence and severity of menopausal symptoms among over 26,000 women in the DACH region, particularly in the psychological and sleep-related domains. Sleep disturbances, unemployment, and weight gain were significantly associated with greater symptom burden, while no clear links were found with lifestyle behaviors such as diet or physical activity. These findings emphasize the need for integrative, targeted support strategies for midlife women. Future research should focus on longitudinal trajectories and interventional approaches, especially addressing sleep and psychosocial risk factors.

## Figures and Tables

**Figure 1 ijerph-22-01502-f001:**
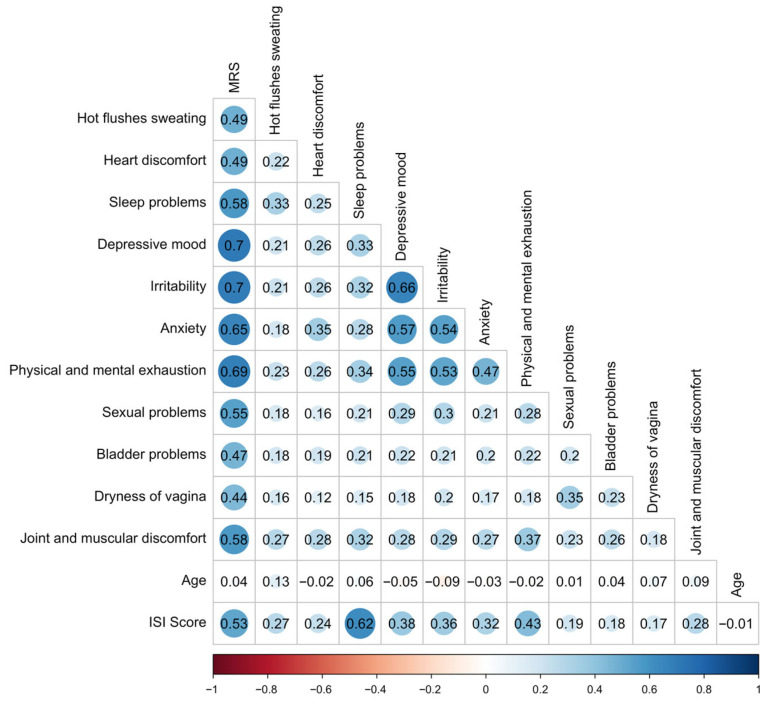
Correlation coefficients r within MRS II items and with age and ISI. The shaded horizontal bar at the bottom of the figure provides a visual scale for interpreting correlation strengths ranging from strong negative (left) to strong positive (right) and corresponds to the classification thresholds used in the manuscript. Correlation strength are as follows: r = ±1 indicates a perfect correlation, ±0.6 to <1 a strong, ±0.3 to <0.6 a weak, and –0.3 to +0.3 no relevant correlation. Positive values reflect direct, negative values inverse associations. Abbreviations: ISI—Insomnia Severity Index, MRS II—Menopause Rating Scale.

**Figure 2 ijerph-22-01502-f002:**
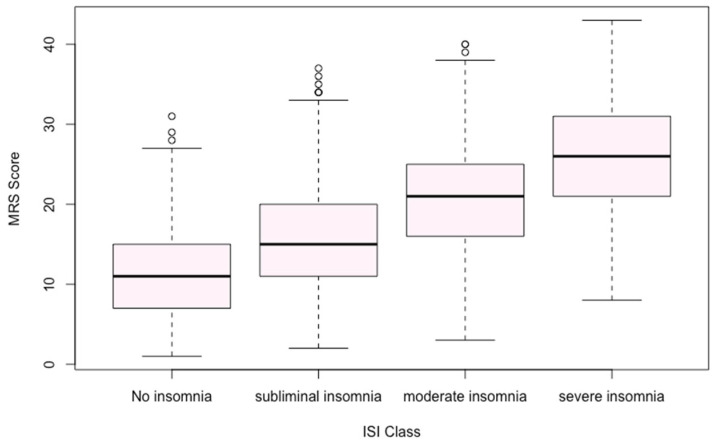
Menopause Rating Scale (MRS II) score and Insomnia Severity Index (ISI) class.

**Figure 3 ijerph-22-01502-f003:**
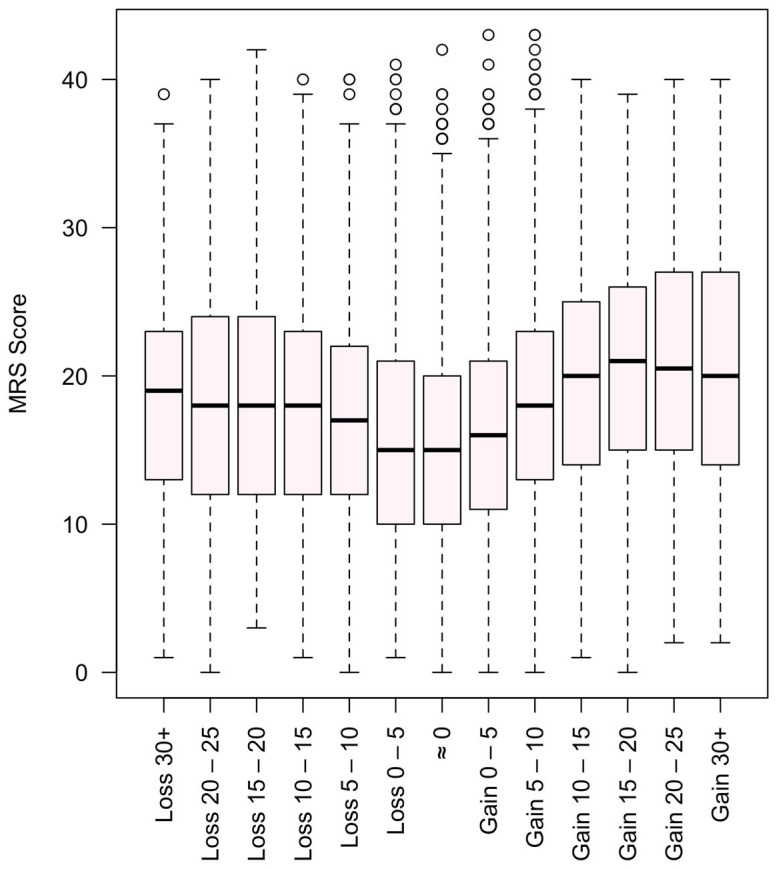
Menopause Rating Scale (MRS II) score and weight gain and loss in kg.

**Table 1 ijerph-22-01502-t001:** Menopause Rating Scale (MRS II).

Psychological Symptoms		Somato-Vegetative Symptoms		Urogenital Symptoms	
**Variable**	**N = 26,338**	**Variable**	**N = 26,338**	**Variable**	**N = 26,338**
**Depressive Mood**		**Hot Flushes Sweating**		**Sexual Problems**	
Mean (SD)	1.79 (1.14)	Mean (SD)	1.31 (1.18)	Mean (SD)	1.59 (1.33)
Median (Q1, Q3)	2.00 (1.00, 3.00)	Median (Q1, Q3)	1.00 (0.00, 2.00)	Median (Q1, Q3)	2.00 (0.00, 3.00)
Min, Max	0.00, 4.00	Min, Max	0.00, 4.00	Min, Max	0.00, 4.00
Unknown	30	Unknown	30	Unknown	30
**Irritability**		**Heart discomfort**		**Bladder problems**	
Mean (SD)	1.90 (1.10)	Mean (SD)	0.99 (1.00)	Mean (SD)	1.04 (1.13)
Median (Q1, Q3)	2.00 (1.00, 3.00)	Median (Q1, Q3)	1.00 (0.00, 2.00)	Median (Q1, Q3)	1.00 (0.00, 2.00)
Min, Max	0.00, 4.00	Min, Max	0.00, 4.00	Min, Max	0.00, 4.00
Unknown	30	Unknown	30	Unknown	30
**Anxiety**		**Sleep problems**		**Dryness of vagina**	
Mean (SD)	1.35 (1.20)	Mean (SD)	2.10 (1.16)	Mean (SD)	0.95 (1.14)
Median (Q1, Q3)	1.00 (0.00, 2.00)	Median (Q1, Q3)	2.00 (1.00, 3.00)	Median (Q1, Q3)	1.00 (0.00, 2.00)
Min, Max	0.00, 4.00	Min, Max	0.00, 4.00	Min, Max	0.00, 4.00
Unknown	30	Unknown	30	Unknown	30
**Physical and mental exhaustion**		**Joint and muscular** **discomfort**		**MRS II** **(complete score)**	
Mean (SD)	2.17 (1.12)	Mean (SD)	1.77 (1.21)	Mean (SD)	16.94 (7.45)
Median (Q1, Q3)	2.00 (1.00, 3.00)	Median (Q1, Q3)	2.00 (1.00, 3.00)	Median (Q1, Q3)	17.00 (11.00, 22.00)
Min, Max	0.00, 4.00	Min, Max	0.00, 4.00	Min, Max	0.00, 44.00
Unknown	30	Unknown	30	Unknown	30

**Table 2 ijerph-22-01502-t002:** MRS II score and ISI class, ^1^ Kruskal–Wallis rank sum test.

Variable	Overall	No Insomnia	Subliminal Insomnia	Moderate Insomnia	Severe Insomnia	*p*-Value ^1^
	N = 4058	N = 393	N = 1692	N = 1716	N = 257	
MRS						<0.001
N Non-missing	4058.00	393.00	1692.00	1716.00	257.00	
Mean (SD)	18.04 (7.35)	11.42 (5.72)	15.78 (6.34)	20.60 (6.51)	25.95 (6.98)	
Median (Q1, Q3)	18.00 (13.00, 23.00)	11.00 (7.00, 15.00)	15.00 (11.00, 20.00)	21.00 (16.00, 25.00)	26.00 (21.00, 31.00)	
Min, Max	1.00, 43.00	1.00, 31.00	2.00, 37.00	3.00, 40.00	8.00, 43.00	

## Data Availability

Data are available from the corresponding author upon reasonable request.

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
