# Peer review of "Women’s Self-Assessment of Quality of Life and Menopausal Symptoms: An Online Survey of 26,000 Women in German-Speaking Countries"

_ijerph, 2025, doi:10.3390/ijerph22101502_

Round 1
Reviewer 1 Report
Comments and Suggestions for Authors
I am grateful for the opportunity to review this article. My comments are as follows:
Abstract:
The abstract provides a general overview of the study. However, it lacks crucial methodological details.
Materials and Methods:
The study relied on social media and online recruitment, which tends to attract women who are already health-conscious, more educated, or experiencing strong symptoms. This introduces selection bias and limits representativeness. The authors should use probability-based or stratified random sampling (e.g., age, socioeconomic status, urban/rural).
The study emphasizes symptoms but does not deeply explore how cultural norms, education, healthcare access, or social support may influence women’s reporting and experiences. The authors should collect socio-demographic and psychosocial data (education level, marital status, cultural attitudes, healthcare-seeking behavior) as well as conduct comparative cross-cultural studies (e.g., between German-speaking countries and Southern Europe or Asia).
Discussion:
While the discussion is strong, it does not fully engage with intersectional factors (e.g., how employment status interacts with education, marital status, or ethnicity). It is suggested to adopt an intersectional analysis framework to better capture how multiple social factors (e.g., class, gender norms, family role) influence symptom burden.
In addition, the study did not sufficiently capture the influence of cultural, socioeconomic, and psychosocial factors on women’s experiences of menopause. A deeper exploration of these dimensions, potentially through the collection of more detailed demographic data and the application of intersectional analytical frameworks, would add valuable insight.
Reference:
To ensure the study remains relevant and reflects current developments in the field, the authors should consider incorporating more recent literature, particularly from the past five years.
Thank you.
Author Response
I am grateful for the opportunity to review this article. My comments are as follows:
Abstract:
The abstract provides a general overview of the study. However, it lacks crucial methodological details.
We thank the reviewer for the valuable comment. We have revised the abstract to include a brief methodological clarification. Specifically, we added the sentence:
“We investigated correlations and group differences using descriptive statistics and univariate tests.”
This addition ensures that the analytical approach is transparently stated within the abstract.
Materials and Methods:
The study relied on social media and online recruitment, which tends to attract women who are already health-conscious, more educated, or experiencing strong symptoms. This introduces selection bias and limits representativeness. The authors should use probability-based or stratified random sampling (e.g., age, socioeconomic status, urban/rural).
The study emphasizes symptoms but does not deeply explore how cultural norms, education, healthcare access, or social support may influence women’s reporting and experiences. The authors should collect socio-demographic and psychosocial data (education level, marital status, cultural attitudes, healthcare-seeking behavior) as well as conduct comparative cross-cultural studies (e.g., between German-speaking countries and Southern Europe or Asia).
We thank the reviewer for raising these important points. We fully acknowledge that our recruitment strategy via widely used social media platforms such as Instagram and Facebook may have introduced selection bias, favoring health-conscious or symptomatic women. This is explicitly addressed in our limitations section as follows:
„… Second, the study may be affected by selection bias, as participation was voluntary and recruitment was conducted exclusively online. This may have skewed the sample toward digitally literate, more educated, health-conscious individuals who are more likely to engage with menopause-related content. Third, self-referral bias cannot be ruled out, i.e. women currently experiencing significant distress or experience strong symptoms may have been more motivated to complete the survey, possibly inflating symptom prevalence estimates….“
While we recognize the value of probability-based or stratified sampling designs, such approaches were beyond the logistical scope and intent of our study, which aimed to generate large-scale, real-world insights from a convenience-based population in the DACH region. Our primary objective was to address the data gap on menopausal experiences in German-speaking countries. As such, we did not aim for a nationally representative sample but rather to highlight symptom burden and associated factors across a broad and voluntary participant base. In terms of sociodemographic detail, we did collect information on age, country of residence, employment status, and weight change. However, we agree that further inclusion of variables such as education level, healthcare access, and cultural norms would enrich the dataset and provide additional context for symptom reporting. These aspects could be the subject of follow-up studies. A smaller sub-cohort (n = 494) provided extended data, including items related to number of children, urban/rural and social support, which we aim to analyze in a separate report.
Regarding the suggestion of cross-cultural comparisons, we agree this represents a valuable future direction. However, our present study was designed specifically to address knowledge gaps within the DACH region. Broader international comparisons were beyond the scope of this analysis but are an important goal for future multicenter research.
Discussion:
While the discussion is strong, it does not fully engage with intersectional factors (e.g., how employment status interacts with education, marital status, or ethnicity). It is suggested to adopt an intersectional analysis framework to better capture how multiple social factors (e.g., class, gender norms, family role) influence symptom burden.
In addition, the study did not sufficiently capture the influence of cultural, socioeconomic, and psychosocial factors on women’s experiences of menopause. A deeper exploration of these dimensions, potentially through the collection of more detailed demographic data and the application of intersectional analytical frameworks, would add valuable insight.
We thank the reviewer for this thoughtful and valuable suggestion. We fully agree that intersectional analyses offer an important lens through which to better understand the diverse experiences of women during menopause. While our main dataset did not include in-depth socio-demographic variables, a smaller subsample of 494 women completed an extended questionnaire that captured several relevant aspects. In this subsample, women reported between 0 and 4 children (median: 2). Regarding residential setting, 25% lived in large cities, 8.7% in urban regions, 34% in small towns, and 32% in rural areas. The vast majority (93%) resided in Germany, with smaller numbers in Austria and Switzerland. Importantly, 95% of respondents reported having at least a small to large social network, while only 5% reported no social connections. This has been added tot he manuscript. However, these findings provide important context on social support and living conditions, even though the sample size does not permit reliable statements about a correlation with the symptoms nor interaction analyses, however but it does describe the women's situation in greater detail.
We acknowledge that a full intersectional analysis such as the interaction between employment, education, family role, or ethnic background would provide valuable insight into how multiple social determinants jointly shape menopausal experiences. We have now included this point explicitly in the limitations and future directions section of the manuscript, emphasizing the need for expanded demographic and psychosocial profiling in future research.
Reference:
To ensure the study remains relevant and reflects current developments in the field, the authors should consider incorporating more recent literature, particularly from the past five years.
We thank the reviewer for this suggestion. Accordingly, we have added relevant literature particularly from the past five years including:
Strelow B, O’Laughlin D, Anderson T, Cyriac J, Buzzard J, Klindworth A. Menopause Decoded: What’s Happening and How to Manage It. J Prim Care Community Health 2024;15:21501319241307460. https://doi.org/10.1177/21501319241307460.
Gibson CJ, Ajmera M, O’Sullivan F, Shiozawa A, Lozano-Ortega G, Badillo EC, et al. A Systematic Review of Anxiety and Depressive Symptoms Among Women Experiencing Vasomotor Symptoms Across Reproductive Stages in the US. Int J Womens Health 2025;17:537–52. https://doi.org/10.2147/IJWH.S491640.
Troìa L, Garassino M, Volpicelli AI, Fornara A, Libretti A, Surico D, et al. Sleep Disturbance and Perimenopause: A Narrative Review. J Clin Med 2025;14:1479. https://doi.org/10.3390/jcm14051479.
Tal JZ, Suh SA, Dowdle CL, Nowakowski S. Treatment of Insomnia, Insomnia Symptoms, and Obstructive Sleep Apnea During and After Menopause: Therapeutic Approaches. Curr Psychiatry Rev 2015;11:63–83. https://doi.org/10.2174/1573400510666140929194848.
Kwon R, Chang Y, Kim Y, Cho Y, Choi HR, Lim G-Y, et al. Alcohol Consumption Patterns and Risk of Early-Onset Vasomotor Symptoms in Premenopausal Women. Nutrients 2022;14:2276. https://doi.org/10.3390/nu14112276.
Thank you.
Reviewer 2 Report
Comments and Suggestions for Authors
This paper is well written and represents important research for women, specifically in the DACH region, but can be helpful for other populations as well. In particularly, I noted that the discussion does a nice job discussing how these findings relate to previous research, particularly in the areas with a the lack of correlation with alcohol use and diet/exercise.
I have a few specific suggestions for improvement:
- More information on the study recruitment and inclusion/exclusion criteria are needed. Was the online survey marketed for those that have menopausal symptoms? Or for those who are post-menopausal? The inclusion is female and at least 18, so it's important to understand if premenopausal women may have taken the survey and skewed the data.
- Related to number 1, please include the age ranges of the population in addition to the mean and SD. Since 43% had a regular period, were they also a lot younger and maybe premenopausal? Or closer to menopausal age? How does that impact your analysis? You stated that "There was no correlation between age and MRS." Could that be because the age range was very tight? Knowing the range is important for interpretation.
- Figure 1 requires more explanation. It's not clear to mean the meaning of the shaded bar at the bottom.
- Throughout all the Figures, make sure that abbreviations are clearly defined and are part of the figure are labeled.
Author Response
This paper is well written and represents important research for women, specifically in the DACH region, but can be helpful for other populations as well. In particularly, I noted that the discussion does a nice job discussing how these findings relate to previous research, particularly in the areas with a the lack of correlation with alcohol use and diet/exercise.
I have a few specific suggestions for improvement:
- More information on the study recruitment and inclusion/exclusion criteria are needed. Was the online survey marketed for those that have menopausal symptoms? Or for those who are post-menopausal? The inclusion is female and at least 18, so it's important to understand if premenopausal women may have taken the survey and skewed the data.
We thank the reviewer for this important observation. We have clarified the recruitment process by adding the following sentence to Section 2.2 Inclusion and Exclusion Criteria:
“The survey was openly advertised as part of a broader initiative on menopause health, primarily via social media channels.”
As such, women experiencing symptoms or with a general interest in the topic may have been more likely to participate. We have acknowledged this potential self-selection bias explicitly in the Limitations section, highlighting how it may have influenced symptom prevalence and generalizability.
- Related to number 1, please include the age ranges of the population in addition to the mean and SD. Since 43% had a regular period, were they also a lot younger and maybe premenopausal? Or closer to menopausal age? How does that impact your analysis? You stated that "There was no correlation between age and MRS." Could that be because the age range was very tight? Knowing the range is important for interpretation.
We thank the reviewer for this thoughtful comment. In response, we have now included the full age range (25 to 75 years) in the Results section. However, a post-hoc analysis revealed that only 36 participants were younger than 40 years, and just 7 were older than 65 years, confirming that the age distribution was indeed narrow in practice. This explains the lack of correlation between age and MRS scores, which we now explicitly clarify in the results and discussion section.
Furthermore, we examined average ages by menstrual status:
- Women with regular periods were 46.19 years old on average
- Women with irregular periods were 47.94 years old
- Women with no periods were 51.31 years old
While these averages show a logical progression in line with menopausal transition, we did not observe a clear corresponding pattern in MRS symptom severity. Furthermore, regardless of their period status, all women were older than 45 years on average. This has been acknowledged in the results and the discussion section of the manuscript to help interpret age-related findings more accurately.
- Figure 1 requires more explanation. It's not clear to mean the meaning of the shaded bar at the bottom.
We thank the reviewer for this valuable observation. We have revised the figure legend to include a clear explanation of the shaded horizontal bar. It now reads as follows:
“The shaded horizontal bar at the bottom of the figure provides a visual scale for interpreting correlation strengths ranging from strong negative (left) to strong positive (right) and corresponds to the classification thresholds used in the manuscript. Correlation strengths are as follows: r = ±1 indicates a perfect correlation, ±0.6 to <1 a strong, ±0.3 to <0.6 a weak, and –0.3 to +0.3 no relevant correlation. Positive values reflect direct, negative values inverse associations.”
We hope this clarifies the visual and conceptual purpose of the figure.
- Throughout all the Figures, make sure that abbreviations are clearly defined and are part of the figure are labeled.
We appreciate the reviewer’s attention to clarity and detail. In response, we have thoroughly reviewed all figures and ensured that all abbreviations are clearly defined either within the figure itself or in the respective figure legends. This includes commonly used terms such as MRS (Menopause Rating Scale), ISI (Insomnia Severity Index), and AUDIT-C (Alcohol Use Disorders Identification Test – Consumption). These changes were made to enhance readability and ensure consistent understanding for all readers.
